# Endemic Circulation of Cluster 19 African Swine Fever Virus in Serbia and Bosnia and Herzegovina

**DOI:** 10.3390/vetsci12111086

**Published:** 2025-11-14

**Authors:** Dimitrije Glišić, Šejla Goletić Imamović, Sofija Šolaja, Ilma Terzić, Ajla Hodžić Borić, Teufik Goletić, Vesna Milicevic

**Affiliations:** 1Department of Virology, Institute of Veterinary Medicine of Serbia, Janisa Janulisa 14, 11000 Belgrade, Serbia; sofija.solaja@nivs.rs (S.Š.); vesna.milicevic@nivs.rs (V.M.); 2Veterinary Faculty, University of Sarajevo, Zmaja od Bosne 90, 71000 Sarajevo, Bosnia and Herzegovina; sejla.goletic@vfs.unsa.ba (Š.G.I.); ilma.terzic@vfs.unsa.ba (I.T.); ajla.hodzic.boric@vfs.unsa.ba (A.H.B.); teufik.goletic@vfs.unsa.ba (T.G.)

**Keywords:** African swine fever, genetic analysis, genotyping, domestic pig, wild boar, Serbia, Bosnia and Herzegovina

## Abstract

African swine fever is a deadly viral disease of pigs and wild boar that causes major losses for farmers and threatens food security. The disease does not affect people, but its rapid spread and high fatality in pigs make it one of the most serious challenges for animal health in Europe. Since 2019, the disease has been present in Serbia, and in 2023, it was first reported in Bosnia and Herzegovina. In this study, we examined virus samples collected from pigs and wild boar during outbreaks between 2023 and 2025 to better understand how the virus is spreading in the region. By looking at several important parts of the virus genome, we found that all the samples belonged to the same group, known as cluster 19. This shows that the same type of virus has been circulating for several years without major changes. The results suggest that the disease is being maintained locally, mainly through contact between wild boar and pigs kept on small farms with little or no protection. The discovery of the same virus type in Bosnia and Herzegovina highlights that the disease crosses borders, making regional cooperation and continued monitoring essential for controlling its spread.

## 1. Introduction

African swine fever (ASF) is a highly fatal viral haemorrhagic disease affecting members of the *Sus scrofa* species. The causative agent, African swine fever virus (ASFV), is the sole member of the *Asfarviridae* family and the *Asfivirus* genus [1]. ASFV has a linear double-stranded DNA genome ranging from 170 to 190 kbp in length, depending on the genotype, subgenotype, and the number of repeat sequences. The genome encodes 150–160 open reading frames, capable of producing over 150 structural and non-structural viral proteins [2]. To date, 24 viral genotypes have been identified, all originating in Africa [2]. However, a recent study showed that genotype XVIII, previously represented by a single Namibian isolate, was a composite of genotypes I and VIII, and its retirement has been recommended to avoid future confusion [3]. Only two genotypes have spread beyond the African continent: genotype I, which caused outbreaks in Europe and the Americas during the 20th century, and genotype II, responsible for the ongoing global pandemic [4]. Genotype I was eradicated from mainland Europe by 1995, but persisted on the island of Sardinia, where it is now considered close to eradication, marking a significant milestone in European ASF control efforts [4].

ASF was first confirmed in Serbia in 2019 in domestic pigs and in 2020 in wild boar [5]. Since then, case numbers have steadily increased in both populations. The highest number of outbreaks was recorded in 2023, with 992 in domestic pigs and 213 in wild boar [6]. In 2024, the numbers dropped to 310 outbreaks in domestic pigs and 101 in wild boar [7]. Bosnia and Herzegovina reported its first case in 2023. That same year, it recorded 1511 outbreaks in domestic pigs and 29 cases in wild boar. In 2024, numbers fell to 33 outbreaks in domestic pigs and 38 in wild boar [6,7]. In 2025, the trend continues downward. Serbia has reported 20 outbreaks in domestic pigs and 28 in wild boar. Bosnia and Herzegovina has reported 8 outbreaks in domestic pigs and 15 in wild boar [8]. In Serbia and Bosnia and Herzegovina, pig farming is largely based on extensive systems [9]. Over 40% of registered farms are smallholdings with low or no biosecurity [10]. Most are family-owned and keep a sow with piglets for personal use. In some regions, pigs are seasonally pastured, increasing contact with wild boar. In other areas, domestic pig–wild boar hybrids are kept and valued as a delicacy, allowing for bidirectional contact in the domestic pig-wild boar interface [11,12]. Since the introduction of ASF, the domestic pig population has decreased significantly. Between 2020 and 2024, the pig population in Serbia declined by approximately 25%, while the European Union (EU-27) experienced a reduction of about 10% over the same period, driven by ASF outbreaks, high production costs, and reduced market demand [13,14,15].

Genotype II ASFV strains in Europe have been divided into over 24 genetic clusters based on multi-gene sequencing, as recommended by the European Reference Laboratory (EURL) [4,16]. Genotyping of ASFV is primarily based on partial sequencing of the B646L gene, which distinguishes 23 known genotypes [3,17]. For deeper classification within genotype II, additional markers are used. The CVR region of the B602L gene, based on the amino acid tandem repeat sequences (TRS), defines 31 subgroups in Africa and two major variants in Europe, with further diversity identified through single-nucleotide polymorphisms (SNPs). Intergenic regions (IGR) also contribute to sub-grouping, particularly the ECO1 region (IGR between I73R and I329L), where four variants (IGR I–IV) are defined by the TRS “TATATAGGAA”, with IGR II being the most common in Europe. The O174L gene allows the distinction of two variants based on the TRS “CAGTAGTGATTTTT”, while TRS in the MGF 505 9R/10R IGR (“AGTAGTTCAGTTAAGAT”) define eight variants, with variant I dominating in European strains. Further differentiation is possible through SNP analysis in the K145R and I215L (ECO2) genes, which define four and two variants, respectively [4]. Based on the multi-gene approach, cluster 19—defined by the markers CVR-1, IGR-II, O174L-I, K145R-I, MGF-I, and ECO2-II—is the dominant variant in Bulgaria, Serbia, Greece, and North Macedonia [4]. In Serbia, additional variations in the CVR and IGR regions have been identified, suggesting the presence of other clusters [16]. In 2023, outbreaks were also reported in Croatia and North Macedonia [6]. Previous spatio-temporal analyses of ASF-infected wild boar showed clusters extending beyond national borders, suggesting regional circulation of related strains [10].

Although ASF has been circulating in Serbia since 2019 and was first detected in Bosnia and Herzegovina in 2023, almost no molecular data have been generated for either country in the past three years. As a result, it is unclear whether the current outbreaks are caused by the continued circulation of previously introduced strains or by new incursions from neighbouring regions. This uncertainty limits veterinary services’ ability to trace transmission routes, evaluate control measures, and detect emerging viral variants. The need for updated genetic data is particularly relevant in the Balkans, where small-scale pig farming, low biosecurity, and frequent cross-border movement of pigs and pork products create favourable conditions for virus persistence. Molecular epidemiology helps determine whether ASFV outbreaks result from new introductions or continued local circulation. Although genotype II ASFV shows minimal genetic variation, sequencing of several genomic regions still allows classification into more than 24 clusters across Europe [4]. The virus evolves slowly, and because cluster 19 predominates in Southeast Europe, tracing transmission routes can be difficult using routine markers. Even so, regions such as the CVR in B602L, the ECO1 and ECO2 intergenic regions, and O174L remain useful for detecting minor differences and distinguishing between persistence and new incursions [4,18]. Beyond the structural gaps in surveillance and biosecurity, everyday human activities play a major role in keeping ASF circulating in the Balkans [9]. The disease often spreads quietly through informal trade of pigs and pork products, home slaughtering, and the delayed reporting of sick or dead animals. This situation is not unique to Serbia and Bosnia and Herzegovina; in Romania and Bulgaria, the highest number of outbreaks has also been recorded in backyard farms with poor biosecurity [19]. In many rural areas, local cross-border movements of pigs, carcasses or equipment take place with minimal control, allowing the virus to move freely between communities and across national borders. Together, these social and economic realities create conditions that allow the virus to persist, even when official outbreak numbers begin to drop. Earlier studies from Serbia reported that genotype II strains belonged mainly to cluster 19, with occasional variation in the CVR and intergenic regions, suggesting microevolution or multiple introductions. This study aims to address the current gap by sequencing ASFV samples from domestic pigs and wild boar collected during outbreaks in Serbia and Bosnia and Herzegovina between 2023 and 2025.

## 2. Materials and Methods

A total of 69 samples from Serbia and 40 samples from Bosnia and Herzegovina were subjected to total nucleic acid extraction, RT-qPCR and subsequent sequencing. Samples from Bosnia and Herzegovina included 34 spleen samples (19 from wild boar and 15 from domestic pigs), three whole blood samples and three serum samples (domestic pigs), while samples from Serbia included 67 blood swabs (domestic pigs) and two spleen samples (wild boar). All samples were collected as part of official ASF surveillance and outbreak control activities, in compliance with national veterinary legislation in Serbia and Bosnia and Herzegovina. As the material was obtained within these compulsory control measures and not for experimental purposes, additional ethical approval was not required [20,21]. Spleen samples, cut into approximately 0.1 g pieces, were homogenized in 1000 μL PBS using a metal bead, while each swab was placed in 1 mL of PBS and homogenized with a TissueLyser II (Qiagen, Hilden, Germany) for 5 min at 30 Hz. The samples were then briefly centrifuged, and the supernatant was used for further extraction steps using the DNeasy Blood & Tissue (Qiagen, Hilden, Germany) according to the manufacturer’s instructions. Whole blood samples and serum samples were extracted using the Purification of Total DNA from Animal Blood or Cells protocol of the DNeasy Blood & Tissue Kit (Qiagen, Hilden, Germany) according to the manufacturer’s instructions. Any remaining tissue samples were stored at −80 °C. Nucleic acid extraction from samples collected in Serbia was performed using the IndiSpin Pathogen Kit (Indical Bioscience GmbH, Leipzig, Germany), following the manufacturer’s protocol. To confirm the presence of the viral RNA, an RT-qPCR protocol by King et al. [22] was used as described by Protocols and Guidelines for Laboratory Diagnosis of ASFV by WOAH [23].

The samples were chosen for sequencing based on their Ct value (≤27) to ensure sufficient template quantity, origin (wild/domestic pig) and the time of sampling to ensure that representative data for outbreaks 2023–2025 would be obtained. As per recommendation by the EURL, targeted sequencing was performed using primers targeting seven regions of ASFV genome: 487 bp region of the B646L locus [17], 491 bp region of the CVR of B602L locus [24] 356 bp TRS present in the IGR between I73R and I329L (ECO1) [25], 637 bp region of O174 locus [26], 282 bp region of K145R locus [26], 551 bp TRS located in the IGR between the MGF505 9R and 10R loci (MGF) [4] and 604 bp fragment containing segments of I329L locus, I215L locus and IGR between them (ECO2) [4].

PCR amplification conditions differed slightly between genomic regions. For the B646L locus, the reaction began with an initial denaturation at 95 °C for 20 s, followed by 40 cycles of 95 °C for 12 s (denaturation), 50 °C for 20 s (annealing), and 70 °C for 25 s (extension). A final extension step was performed at 72 °C for 10 min. For the ECO1 locus, amplification was carried out using 40 cycles of 95 °C for 30 s, 60 °C for 30 s, and 72 °C for 1 min, with a final extension at 72 °C for 10 min. For the B602L, MGF, and ECO2 regions, the cycling parameters were identical, except that the annealing temperatures were 55 °C for B602L and MGF, and 56 °C for ECO2. For the K145R and O174L loci, PCR conditions followed previously published protocols and consisted of an initial denaturation at 95 °C for 10 min, followed by 40 cycles of denaturation at 95 °C for 15 s, annealing at 50 °C for 30 s, and extension at 72 °C for 30 s. A final elongation was performed at 72 °C for 3 min [17,24,25,26]. The quantification of DNA was performed using a Qubit dsDNA HS (High Sensitivity) Assay Kit on a Qubit 4 device (Thermo Fisher Scientific, Waltham, MA, USA), following the manufacturers’ instructions. For samples from Bosnia and Herzegovina sequencing of the obtained amplicons was performed using Native Barcoding Kit 24 V14 (SQK-NBD114.24, Oxford Nanopore Technologies, Oxford, UK) according to the manufacturer’s instructions, on an R10.4.1 flow cell using the MinION Mk1C instrument (Oxford Nanopore Technologies, Oxford, UK). Sequences were basecalled in real time through MinKNOW v.20.10.3 using the high accuracy setting and a quality score set to 9 during the 12 h runtime in order to obtain high-quality coverage of the sequences. Seqkit v.2.8.2 (accessed on the 22 September 2025 https://bioinf.shenwei.me/seqkit/) was used to additionally check sequence quality, after which the reads were mapped to their respective loci on the Georgia 2007/1 reference genome (GenBank No.: FR682468.2) with Minimap2 v2.29-r1283 (accessed on the 22 September 2025 https://github.com/lh3/minimap2/releases). The sequences were then checked and indexed using Samtools 1.21 (accessed on the 22 September 2025 https://github.com/samtools/samtools/releases/), mappings were visually analyzed using UGENE v52.0 [27] and consensus sequences were created with iVar v1.4.3 (accessed on the 22 September 2025 https://github.com/andersen-lab/ivar). PCR products obtained from Serbian samples were purified using the GeneJET PCR Purification Kit (Thermo Fisher Scientific, Waltham, MA, USA) and subsequently sequenced by Sanger sequencing at LGC, Biosearch Technologies (Berlin, Germany). The obtained sequences were run against the NCBI database with the BLAST tool 2.17.0 (accessed on the 22 September 2025 https://blast.ncbi.nlm.nih.gov/Blast.cgi) to confirm genetic similarities with other ASFV sequences and determine their genotype in accordance with EURL standards. All sequences were deposited in GenBank (Appendix A).

Finally, the epidemiological data of sequenced samples were used to generate the map of the localities with ASF cases in domestic pigs and wild boar.

## 3. Results

### Sequencing Success and Genetic Classification of ASFV Isolates

From the 69 ASFV isolates originating from Serbia and selected for sequencing, all were successfully amplified on gel PCR; however, good-quality nucleotide sequences were obtained for 67 isolates at the B646L locus (97.1%), 59 at the B602L locus (85.5%), 61 in the ECO I region (88.4%), 68 in the ECO II region (98.6%), 54 at the O174L locus (78.3%), 65 at the K145R locus (94.2%), and 64 in the MGF region (92.8%). Each isolate yielded at least two genomic regions with reliable sequence data, allowing their inclusion in downstream comparative analyses (Appendix A). All 40 ASFV isolates from Bosnia and Herzegovina yielded good-quality sequences of all seven targeted regions, and as such, all were used for downstream analysis. Alignment of the B646L gene confirmed that all isolates belonged to genotype II. Further classification followed the EURL recommendations. Analysis of the CVR amino acid TRS (BNDBNDBNAA) revealed 100% identity with the reference strain Georgia 2007/1 (FR682468.1), classifying all isolates as CVR variant I. Examination of the TRS “TATATAGGAA” in the ECO1 region classified all isolates as IGR-II, while analysis of the O174L TRS placed them within O174L-I. Similarly, the TRS within the MGF 505 9R/10R IGR (“AGTAGTTCAGTTAAGAT”) classified the isolates as MGF-I. SNP analysis in the K145R and I215L genes further grouped the isolates as K145R-I and ECO2-II, respectively. According to the EURL guidelines, these combined markers place all strains sequenced between 2023 and 2025 into cluster 19.

The spatial distribution of ASF-positive cases in domestic pigs and wild boar in Serbia and Bosnia and Herzegovina over the 2023–2025 period is shown in Figure 1.

## 4. Discussion

This study provides the first updated genetic characterization of ASFV strains circulating in Serbia during 2023–2025 and, importantly, presents the first sequence data from Bosnia and Herzegovina.

The results confirm the continued dominance of cluster 19 in the Balkan region, consistent with earlier reports, and underline the need for sustained molecular surveillance to monitor ASFV dynamics [4]. Elsewhere in Europe, greater diversity has been reported. Romania, for example, shows the highest level of genetic variability, with Gallardo et al. (2023) identifying six distinct ASFV clusters, some apparently unique to that country [4]. At a broader European scale, the highest variability has been associated with the O174L-II variant, which is found in Romania, Poland, and Germany, allowing for fine regional tracking of different strains [4,18]. This variant is associated with changes in the PolX DNA polymerase, which may contribute to increased mutation rates, as well as allowing for regional mapping of strains [18]. By contrast, our findings from Serbia and Bosnia and Herzegovina revealed no such variability; all sequenced isolates were classified as cluster 19, suggesting a genetically stable viral population over the study period. Previous investigations in Serbia had demonstrated a more complex scene. Both ECO2-I and ECO2-II variants were detected, with ECO2-I linked to a secondary introduction of the virus and ECO2-II to the initial incursion [28]. These results were interpreted as evidence of at least two independent introductions, possibly from Romania (ECO2-I) and Bulgaria (ECO2-II). Other studies also identified variation in the ECO1 region and in the CVR of the B602L gene, including the first detection of ECO1 IGR-III and several CVR-specific SNP variants [16]. In the present dataset, none of these were observed. The most likely explanation is that such variants either faded out without wider spread or represented restricted circulation events that were only captured incidentally. Similar findings have been described in the Baltic countries, where ASFV has persisted for more than a decade without the emergence of the high number of clusters reported in neighbouring Poland [4].

No novel genetic changes were observed in this study, reinforcing the view that cluster 19 remains the only strain circulating in Serbia and Bosnia and Herzegovina. It should be noted that the statement “no new genetic changes were observed” refers exclusively to the seven genomic regions analyzed. The study did not employ whole-genome sequencing; therefore, mutations outside these loci could not be assessed, and genome completeness or viral infectivity cannot be inferred from the present data. ASFV has now been established in both domestic pigs and wild boar in Serbia for more than five years and can be considered endemic. Although case numbers have declined since 2023, sporadic outbreaks continue to be reported in both countries, indicating that the virus remains present in local populations. Transmission is most likely sustained through interactions at the wild boar–human–domestic pig interface, particularly in backyard farms where biosecurity is minimal or absent. Practices such as unregulated pig movements (e.g., unregulated trade) and the illegal disposal of carcasses further maintain infection pressure in wild boar populations. The persistence of a single stable cluster over several years suggests that once established, ASFV can be maintained without repeated introductions, largely through ecological and human-mediated cycles. Although our molecular findings confirm the continued circulation of a single ASFV cluster, they alone cannot explain how the virus spreads. Considering the close cultural and trade connections between Serbia and Bosnia and Herzegovina, the movement of pigs and pig products across the border likely plays a role in maintaining transmission.

In wild boar populations, ASFV can be maintained independently of domestic pigs, primarily through carcass-mediated transmission. Infected carcasses act as long-term sources of infection, and although their detection and removal are mandated, reporting by hunters and foresters is often inconsistent or delayed, as it may not align with their personal or economic interests [29]. Experimental and field data show that ASFV remains viable in bone marrow and lymphatic tissues for weeks to months, especially in colder weather or shaded forest environments. Field observations show that wild boar frequently investigate and return to carcasses of conspecifics, which increases the likelihood of contact with infected tissue or contaminated surroundings [30]. In stakeholder surveys, hunters and foresters reported that carcass reporting and removal are sometimes delayed or inconsistently performed due to lack of incentives and practical constraints [29]. In combination with practices such as supplementary feeding or aggregation, these factors contribute to sustained circulation of ASFV within wild boar populations, even in the absence of domestic pig involvement [30]. If carcasses are not located and removed, healthy wild boar may become infected while scavenging or through contaminated soil and vegetation [30]. Practices such as supplementary feeding, delayed carcass disposal and uncontrolled movement of wild boar across administrative borders further support this cycle of persistence. These ecological conditions help explain why ASF continues to circulate even when genetic analyses suggest a stable, unchanged viral population. Future studies that combine molecular data with field investigations of farming practices and animal trade would help clarify these dynamics more fully. As several analyzed loci, including ECO1 and the MGF 505 IGR, contain repetitive sequences that hinder reliable alignment, and given the >99.9% sequence identity reported among genotype II ASFV isolates across Europe, phylogenetic analysis was not conducted, as it would not yield meaningful resolution beyond cluster assignment.

The situation in Bosnia and Herzegovina illustrates the transboundary nature of ASFV spread. The first case in domestic pigs was reported in 2023 in the Republic of Srpska, bordering Serbia [6]. Even though the ASF outbreaks in domestic pigs in Bosnia and Herzegovina remain focused around several localities, the affected wild boar are present throughout the central and eastern parts of the country. This likely enables the spread of the virus between localities both in the country and between two countries (Figure 1). Additionally, many cases likely remain unreported, and the number of positive cases is likely higher. The detection of cluster 19 in both countries provides strong evidence of cross-border circulation and emphasizes the importance of regional cooperation. In Serbia and Bosnia and Herzegovina, ASF control is based on the immediate culling of affected backyard herds, strict movement restrictions, and active carcass collection and disposal, together with efforts to improve farm biosecurity. These measures largely explain the absence of pathological findings and prevent any reliable estimation of natural mortality in the present study [20,21]. Without coordinated surveillance and control, the virus will continue to move across national borders, undermining individual countries’ efforts. This study is limited by the sequencing of only a subset of outbreaks and by the lower resolution of targeted sequencing compared to whole-genome approaches. The cost-effectiveness of this sequencing approach is questionable, as sequencing seven genome fragments per sample is labour-intensive and often yields largely identical sequences that provide limited regional resolution. Still, mutations in certain loci, such as the O174L segment noted earlier, may offer valuable discriminatory power for more precise regional tracking of circulating strains. Identifying which genome segments are most informative for ASFV, however, remains challenging. Whole-genome sequencing would, in principle, provide the most reliable insight, yet it is both costly and labour-intensive, and in practice, contributes little additional information given that most genotype II ASFV genomes share over 99.9% identity [31]. Additionally, whole genome sequencing would enable tracking the evolution of the entire genome of ASFV, and this may especially be of great importance in the context of persistence of ASFV in the population of wild boar. This approach would also enable the determination of the origin of outbreaks in domestic pigs. Future work should expand outbreak sampling and incorporate whole-genome sequencing to capture finer-scale variation and to improve detection of potential introductions of novel strains. While both Sanger and targeted sequencing lack the resolution of next-generation approaches, they remain useful for cluster assignment and for confirming the absence of new variants during the study period. In our case, even though the same primers were used, Sanger sequencing yielded incomplete coverage across several regions, whereas the targeted MinION approach produced consistent results throughout, which may indicate a greater robustness of nanopore sequencing to template quality and reaction variability. Occasional sequencing failures in the Serbian set likely arose from reduced template quality or DNA degradation during transport, rather than from genomic deletions, as the targeted ASFV loci are essential and highly conserved. Overall, our findings demonstrate the continued dominance of cluster 19 in Serbia and Bosnia and Herzegovina during 2023–2025. Sustained molecular surveillance, combined with cross-border epidemiological coordination, will be essential for tracking ASFV evolution in the region and for informing effective control and eradication strategies.

## 5. Conclusions

This study provides updated genetic data on African swine fever virus from Serbia and the first sequencing results from Bosnia and Herzegovina. All analyzed isolates belonged to cluster 19, confirming its continued dominance and stable circulation in the region between 2023 and 2025. The absence of new variants suggests limited introductions from outside and points to endemic maintenance through the wild boar–domestic pig interface. The detection of the same cluster in both countries highlights the cross-border nature of the disease and the need for ongoing molecular surveillance and regional cooperation. However, our results also show the limits of the seven-fragment sequencing approach: in a genetically stable setting such as cluster 19, this method does not provide enough resolution to track individual isolates or local transmission routes. In these cases, whole-genome sequencing or markers with greater variability are needed to gain more detailed epidemiological insights. In addition, strengthening data sharing between laboratories in the region, alongside harmonized sampling strategies for both domestic pigs and wild boar, would improve the early detection of any emerging variants. A more systematic inclusion of wild boar carcasses, particularly in areas with declining domestic outbreaks, may help clarify whether virus circulation is decreasing or simply shifting further into wildlife reservoirs. Such approaches would support earlier detection of virus reintroduction or silent persistence and help guide more targeted control measures in domestic and wild populations

## Figures and Tables

**Figure 1 vetsci-12-01086-f001:**
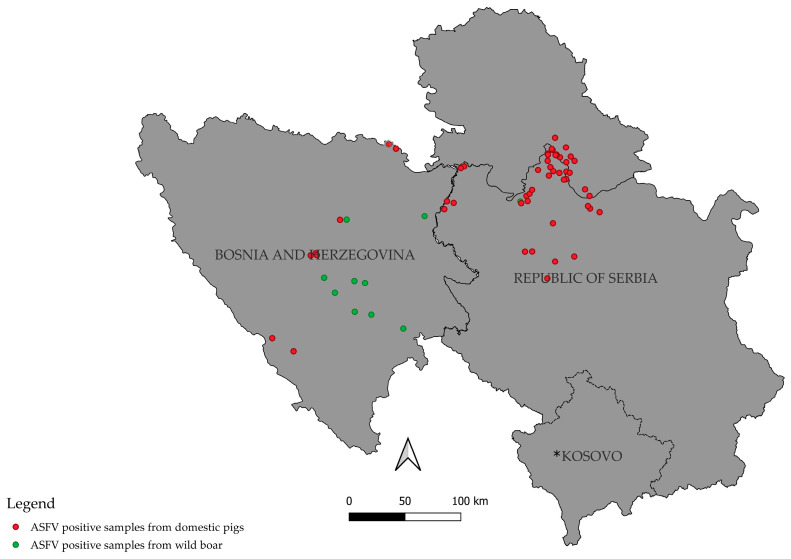
Geographic map of the Republic of Serbia and Bosnia and Herzegovina. Red dots indicate domestic pig samples positive for ASFV, while green dots represent wild boar samples positive for ASFV. * This designation is without prejudice to positions on status and is in accordance with United Nations Security Council Resolution 1244 (1999) and the advisory opinion of the International Court of Justice on the Kosovo Declaration of Independence.

## Data Availability

The original contributions presented in this study are included in the article/Appendix A. Further inquiries can be directed to the corresponding author.

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
