# Peer review of "Endemic Circulation of Cluster 19 African Swine Fever Virus in Serbia and Bosnia and Herzegovina"

_vetsci, 2025, doi:10.3390/vetsci12111086_

Round 1

Reviewer 1 Report

Comments and Suggestions for Authors

The manuscript "Endemic Circulation of Cluster 19 African Swine Fever Virus in 2 Serbia and Bosnia and Herzegovina" detected viral samples collected from domestic pigs and wild boars during the 2023–2025 epidemic period. Results showed that all samples belonged to ASFV Genogroup 19, with no significant sequence changes observed over years of transmission. The epidemic in this region was mainly concentrated in wild boars and small-scale farms without protective measures. The study also described that the viruses endemic in regions of Bosnia and Herzegovina were identical, highlighting the characteristics of the epidemic. However, most of the work in this study is descriptive and lacks necessary evidence for verification.

Majors:

1.Among 69 sequenced samples from Serbia, the authors sequenced 7 key gene segments. Results showed that the B646L gene was amplified in 67 samples, the B602L gene in 59 samples, and the O174L gene in 54 samples. These genes are all key essential genes of ASFV. Why could the corresponding genes not be amplified in some samples? Is it due to incomplete viral strains or problems with sample extraction?

2.The manuscript states that "no new genetic changes were observed in this study". As described in the first question, the identification of ASFV-infected samples can be verified using methods recommended by the WOAH. However, nucleic acid extraction and amplification alone cannot confirm whether the viral genome in the sample is complete or whether the virus is infectious. As the authors stated in the discussion, "whole-genome sequencing or markers with greater variability are required to obtain more detailed epidemiological insights". The manuscript lacks necessary data to verify gene integrity.

Minors:

1."WOA" should be corrected to "WOAH" (full English name: World Organization for Animal Health).

2.The phrase "allowing for food regional tracking of different strains" is presumed to be the author’s intended expression of "allowing for fine regional tracking of different strains" 3."MiniON" should be corrected to "MinION" (Note: Standard product name of the portable nanopore sequencing platform developed by Oxford Nanopore Technologies).

4."High fatality in pigs" should be corrected to "high fatality rate in pigs"

Author Response

Dear Reviewer,

Thank you very much for taking the time to review our manuscript and for providing valuable feedback and suggestions. We carefully considered all comments and have addressed each point in detail below. Corresponding revisions and corrections have been made in the resubmitted manuscript and are highlighted using track changes. We believe that these modifications have improved the clarity, accuracy, and overall quality of the paper, and we sincerely appreciate the reviewers’ and editors’ efforts in this process.

A detailed description of the changes made can be found in the attached PDF.

Kind regards.

Reviewer 2 Report

Comments and Suggestions for Authors

Reviewer comments

The authors in the manuscript entitled “Endemic Circulation of Cluster 19 African Swine Fever Virus in Serbia and Bosnia and Herzegovina” describes about the prevalence of AFSV in Serbia and Bosnia and Herzegovina among pigs and wild boars by amplification of partial gene segments, sequencing and nucleotide GenBank submissions. All the ASFV strains belong to Cluster 19 of Genotype II.

The reviewer has following comments to make-

  • The authors must perform phylogenetic reconstruction using suitable bioinformatics software to understand virus evolution. The authors must include representative strains from different clusters of Genotype II and few strains from Genotype I; strains from other European countries.
  • As the manuscript is an epidemiological study, the authors must describe the number of ASF outbreaks in Serbia and Bosnia and Herzegovina, the number of domestic pigs/ farms and wild boars infected, mortality in the outbreak, pathological lesions (gross and histopathology), possible route of incursion of ASFV from neighbouring countries. Are the outbreaks still continuing and what are the control measures implemented to prevent future outbreaks.
Comments on the Quality of English Language

Can be improved. 

Author Response

Dear Reviewer,

Thank you very much for taking the time to review our manuscript and for providing valuable feedback and suggestions. We carefully considered all comments and have addressed each point in detail below. Corresponding revisions and corrections have been made in the resubmitted manuscript and are highlighted using track changes. We believe that these modifications have improved the clarity, accuracy, and overall quality of the paper, and we sincerely appreciate the reviewers’ and editors’ efforts in this process.

The detailed responses can be found in the attached PDF.

Kind regards.

Reviewer 3 Report

Comments and Suggestions for Authors

The manuscript is well written and provides interesting information about the current circulation of African swine fever virus in Serbia and Bosnia and Herzegovina, as well as identifying possible limitations of the approach used to determine the relationship between the viral isolates included in the study.

While transmission at the wild boar-domestic pig interface clearly occurs and is likely important in maintaining virus circulation, virological studies including in-depth molecular studies using full genome sequencing do not provide all the information needed to understand transmission of the virus. Given the strong cultural ties between the two countries and the fact that first outbreaks in Bosnia and Herzegovina occurred in the Serbian region of the country, a study of the pig value chains in combination with the virological information would most likely identify movement of domestic pigs and pig products across the border as having an important role in maintaining viral circulation. Many studies have shown that wild boar movements over long distances are not usual and are the result of human activity, which should also be investigated. Some information on circumstances that lead to mingling of domestic pigs and wild boars is provided, but in my opinion the importance of molecular genetics in tracing outbreaks is over-rated because it can only indicate the relationship between isolates but rarely the direction in which movement occurred and never the reason why it occurred. A classic example is the introduction of ASFV in the Republic of Georgia in 2007 – molecular genetics indicated that the virus originated in eastern Africa, with greatest similarity to an isolate from Mozambique (Mthombeni et al., 2023), but only information on shipping routes and the provisioning of ships would be able to indicate how such an introduction might have occurred and would be informative for developing prevention and control strategies.

Line 32,34, and elsewhere, for example in the first paragraph of Results: Replace ‘strains’ with ‘isolates’. The term strains generally refers to groups of isolates that are genetically different from others, so, as indicated in lines 215-216, with Cluster 19 you are in fact dealing with a single strain of ASFV.

Line 81 and elsewhere: A recent publication described the retirement of Genotype XVIII, which was represented by a single isolate from Namibia and with full genome sequencing it was shown to be a composite of genotype I (the only genotype so far confirmed in Namibia) and genotype VIII, which is suggestive of cross-contamination in the laboratory (the authors were too polite to mention probable cause). The article recommended retirement of genotype XVIII with all the other 23 genotypes retaining their original numbers to avoid future confusion. Although this issue is not relevant to the situation in Serbia and Bosnia and Herzegovina, it is worth mentioning it both to demonstrate up-to-date knowledge of the literature and to help with dissemination of that information. The article is the following: Goatley, L.C., Freimanis, G.L., Tennakoon, C., Bastos, A., Heath, L., Netherton, C.L. (2024). African swine fever virus NAMP1/95 is a mixture of genotype I and genotype VIII viruses. Microbiology Resource Announcements, 13, 4 doi: 10.1128/mra.00067-24.

Line 123: WOA should be WOAH.

Line 198: ‘food regional tracking’ – should this read ‘good regional tracking’?

Author Response

Dear Reviewer,

Thank you very much for taking the time to review our manuscript and for providing valuable feedback and suggestions. We carefully considered all comments and have addressed each point in detail below. Corresponding revisions and corrections have been made in the resubmitted manuscript and are highlighted using track changes. We believe that these modifications have improved the clarity, accuracy, and overall quality of the paper, and we sincerely appreciate the reviewers’ and editors’ efforts in this process.

The detailed responses can be found in the uploaded PDF.

Kind regards.

Round 2

Reviewer 1 Report

Comments and Suggestions for Authors

Accept for this version

Author Response

Dear reviewer,

Thank you very much for taking the time to review our manuscript and for providing valuable feedback and suggestions. 

Kind regards.

Reviewer 2 Report

Comments and Suggestions for Authors

The reviewer agrees with the authors comments regarding the queries raised. The authors must justify why they have taken 7 targets, it the same genotyping can of ASFV can be done with 2-3 targets. As the study does not add up substantial information in the field of ASFV epidemiology, why this article requires a full length article and not a short or brief communication? 

Comments on the Quality of English Language

Can be improved. 

Author Response

Dear Reviewer,

Thank you very much for taking the time to review our manuscript and for providing valuable feedback and suggestions. We carefully considered all comments and have addressed each point in detail below. Corresponding revisions and corrections have been made in the resubmitted manuscript and are highlighted using track changes. We believe that these modifications have improved the clarity, accuracy, and overall quality of the paper, and we sincerely appreciate the reviewers’ and editors’ efforts in this process. Additional comments can be found in the attachment.

Kind regards.
